# Clinical Results of Fibroblast Activation Protein (FAP) Specific PET and Implications for Radiotherapy Planning: Systematic Review

**DOI:** 10.3390/cancers12092629

**Published:** 2020-09-15

**Authors:** Paul Windisch, Daniel R. Zwahlen, Stefan A. Koerber, Frederik L. Giesel, Jürgen Debus, Uwe Haberkorn, Sebastian Adeberg

**Affiliations:** 1Department of Radiation Oncology, Kantonsspital Winterthur, 8400 Winterthur, Switzerland; daniel.zwahlen@ksw.ch; 2Department of Radiation Oncology, Heidelberg University Hospital, 69120 Heidelberg, Germany; Stefan.Koerber@med.uni-heidelberg.de (S.A.K.); juergen.debus@med.uni-heidelberg.de (J.D.); Sebastian.Adeberg@med.uni-heidelberg.de (S.A.); 3Heidelberg Institute of Radiation Oncology (HIRO), 69120 Heidelberg, Germany; 4National Center for Tumor Diseases (NCT), 69120 Heidelberg, Germany; 5Clinical Cooperation Unit Radiation Oncology, German Cancer Research Center (DKFZ), 69120 Heidelberg, Germany; 6Heidelberg Ion-Beam Therapy Center (HIT), Department of Radiation Oncology, Heidelberg University Hospital, 69120 Heidelberg, Germany; 7German Cancer Consortium (DKTK), 69120 Heidelberg, Germany; 8Department of Nuclear Medicine, Heidelberg University Hospital, 69120 Heidelberg, Germany; frederik.giesel@med.uni-heidelberg.de (F.L.G.); uwe.haberkorn@med.uni-heidelberg.de (U.H.); 9Clinical Cooperation Unit Nuclear Medicine, German Cancer Research Center (DKFZ), 69120 Heidelberg, Germany; 10Translational Lung Research Center Heidelberg (TLRC), German Center for Lung Research (DZL), 69120 Heidelberg, Germany

**Keywords:** fibroblast activation protein, positron emission tomography, radiotherapy, staging, positron emission tomography-computed tomography, PET, PET-CT

## Abstract

**Simple Summary:**

The purpose of this systematic review is to summarize the findings of using fibroblast activation protein (FAP) as a new tracer for positron emission tomography (PET) in cancer patients. In the 19 included studies, FAP was evaluated in various cancers and showed promising initial results for staging as well as radiotherapy planning.

**Abstract:**

Small molecules targeting fibroblast activation protein (FAP) have emerged as a new group of tracers for positron emission tomography (PET) in 2018. The purpose of this systematic review is therefore to summarize the evidence that has been gathered to date in patients and to discuss its possible implications for radiotherapy planning. The MEDLINE database was searched for the use of FAP-specific PET in cancer patients and the records were screened according to PRISMA guidelines. Nineteen studies were included. While dedicated analyses of FAP-specific PET for radiotherapy planning were available for glioblastoma, head and neck cancers, lung cancer, and tumors of the lower gastrointestinal tract, there is still very limited data for several epidemiologically significant cancers. In conclusion, FAP-specific PET represents a promising imaging modality for radiotherapy planning that warrants further research.

## 1. Introduction

Target volume delineation has always been but has become increasingly challenging in radiation oncology given the ever more prevalent use of techniques that enable highly conformal radiotherapies such as intensity-modulated or stereotactic radiation therapy. 

As dose gradients become steeper, parts of the tumor which cannot be detected by pretreatment imaging may not be included and receive sufficient dose as the supposedly normal tissue in the vicinity of the tumor is spared.

Developing new techniques to complement established imaging modalities such as computed tomography (CT) or magnetic resonance imaging (MRI) with or without contrast enhancement is therefore of great importance not just from a diagnostic but also a radiation oncology perspective. 

One modality which has provided continuous innovation and therefore seen increasing use has been the positron emission tomography-computed tomography (PET-CT), mainly due to the development of new tracers in addition to the most prevalent ^18^F-fluorodeoxyglucose (^18^F-FDG) [1]. PET-CTs are nowadays recommended for a variety of clinical indications and have been shown to alter the management of oncology patients in many cases [2,3].

A novel group of tracers that has emerged recently are substances targeting the fibroblast activation protein (FAP) on the surface of fibroblasts in the tumor stroma, so-called cancer-associated fibroblasts (CAFs) first published in 2018 [4]. As FAP is highly specific to a large subset of CAFs, early molecules targeting FAP have initially been designed to inhibit FAP and to thereby serve as potential therapeutic agents before being used as a PET tracer in the past [5,6]. Even though fibroblasts occur ubiquitously throughout the body, they normally express dipeptidyl peptidase 4 (DPP4) instead of the related FAP, which is why in preclinical in vitro and in vivo studies, FAP-specific tracers showed high specificity, affinity, and rapid internalization [4]. 

Following the first in human publications on FAP-specific PET in heterogeneous patient cohorts with different tumors, studies with a narrower scope have been published, with some also investigating the impact of FAP-specific PET on radiation oncology treatment planning [4,7,8].

As the development of various tracers such as ^68^Ga-prostate-specific membrane antigen (PSMA) has had a considerable impact on radiation oncology and current oncological guidelines, especially in combination with concepts such as oligometastases that have increased the importance of determining the exact spread of a patient’s cancer, herein, we conducted a systematic review of the available data regarding FAP-specific PET and focus on its potential use for radiotherapy treatment planning [9,10]. 

## 2. Results

The inclusion workflow is depicted in Figure 1. The query returned 30 publications and no duplicates. During the screening of the records, six results were excluded due to being only a dataset [11], being a reply [12], presenting results of FAP-specific PET for IgG4-related disease [13,14,15], and focusing on FAP as a way to image cardiotoxicity [16]. During the screening of the full-text articles, 5 articles were excluded due to presenting a patient with tuberculosis [17], presenting solely preclinical results [18], being a letter to the editor [19,20] and presenting a patient with benign liver nodules [21]. Ultimately, 19 articles were included whose characteristics are depicted in Table 1. 

### 2.1. Sources of Bias

A patent application for quinoline-based FAP targeting agents for imaging and therapy in nuclear medicine was the most frequent COI and present in ten publications (53%). The most frequent funding source was the National Natural Science Foundation of China in three publications (16%).

### 2.2. Brain

Three studies describe results of FAP-specific PET performed on the same group of glioma patients [23,34,35]. The first analysis by Röhrich et al. included 13 glioblastomas (GBMs), one grade III and three grade II gliomas. One of the GBMs and all grade III and II gliomas harbored a mutation in the Isocitrate dehydrogenase (IDH) gene. While grade II, IDH mutant gliomas showed only slight tracer uptake and low tumor-to-background ratios, grade III gliomas and GBMs showed stronger enhancement [34]. A second paper by Röhrich et al. analyzed the 13 IDH-wildtype GBMs and found that FAP enhancement was not independent from but not limited to reflecting perfusion differences. In addition, intratumoral differences in enhancement most likely do not reflect cell density but rather differences in FAP expression [35]. A paper by Windisch et al. analyzed the same subset of patients for radiotherapy planning. It observed that applying a 7-fold threshold of intensity compared to healthy-appearing normal brain tissue resulted in PET-based gross tumor volumes (PET-GTVs) equal to those based on MRI (MRI-GTVs). MRI- and PET-GTVs were incongruent, and adding PET- to MRI-GTVs resulted in an increase of the MRI-GTV by on average 45.9% [23].

Chen et al. performed a comparison of FAP-specific to FDG PET that contained 4 glioma patients (2 GBM, 1 grade II glioma, 1 grade III glioma) and noted that while the absolute uptake was lower for FAP-specific PET, the tumor-to-background ratio was higher [12]. 

The authors also report that FAP-specific PET detected more brain metastases, although the number of cases was limited (6 patients with FAP-specific PET vs. 3 patients with FDG PET). Other studies report on the detection of brain metastases with FAP-specific PET as well and also noting that the lower absolute uptake but higher tumor-to-background ratio observed in glioma also applies there [25,38].

### 2.3. Head and Neck

Syed et al. assessed the use of FAP-specific PET for radiotherapy planning in 14 head and neck patients (12 squamous cell carcinoma, 1 mucoepidermoid carcinoma, 1 undifferentiated) and came to the interdisciplinary consensus that a 3-fold threshold compared to healthy-appearing normal tissue was best for target volume delineation in this group. All higher thresholds resulted in volumes significantly smaller compared to conventional target volume delineation with contrast-enhanced CT and MRI [8]. 

Other publications also reported on FAP-specific PET for head and neck tumors, among them the discovery of an occult nasopharyngeal carcinoma in a patient with cancer of an unknown primary (CUP) [25]. It is also mentioned that the low uptake of healthy brain tissue helps determine the tumor spread towards the skull base [36].

### 2.4. Lung

Giesel & Adeberg et al. used a newer compound, FAPI-74, for target volume delineation in 10 patients with non-small-cell lung cancer (NSCLC, 8 adenocarcinomas, 2 squamous cell carcinomas, Figure 2) [24]. Applying a 3-fold threshold compared to healthy-appearing normal tissue resulted in volumes equal to those created by CT-based contouring (69.8 mL with FAP-specific PET vs. 67.4 mL with CT). FAP-specific PET was able to identify additional metastases in a patient previously considered oligometastatic, but the number of cases was too low for the computation of metrics such as sensitivity and specificity. Other publications report the propensity of FAP-specific PET to detect more NSCLC metastases than FDG PET, but patient collectives were too small to analyze whether this translates to more accurate staging.

### 2.5. Breast

Data on FAP-specific PET is still limited. While several publications contain at least one patient with breast cancer, no dedicated analysis for target volume delineation has been performed [12,26,32,36,37]. The biggest collective consisted of four patients (3 invasive ductal carcinomas, 1 invasive lobular carcinoma) analyzed by Chen et al. who found that FAP-specific PET was able to detect more lymph node metastases than FDG PET in this small sample [25]. 

### 2.6. Upper Gastrointestinal Tract

For esophageal cancer, the biggest patient collective was analyzed by Chen et al. and featured 5 squamous cell carcinomas [12]. In a publication by the same group on patients with inconclusive FDG PETs, FAP-specific PET detected an occult esophageal primary in a patient whose cancer was formerly classified as CUP. In two other cases, the tumor stage was changed by detecting additional lymphatic metastases in two patients with esophageal cancer [25]. 

The same publication also reports on 12 patients with gastric cancer (8 adenocarcinoma, 4 signet ring cell carcinoma) and found a significantly higher uptake compared to FDG. 

A case report by Pang et al. used FAP-specific PET to detect signet ring cell carcinoma in a patient formerly treated for prostate cancer. In addition to the detection of the primary and several metastases, the authors report the presence of bilateral adrenal enhancement that they think might be caused by hormonotherapy-induced chronic inflammation [27]. In a case report by Wang et al., FAP-specific PET was used in a patient with gastric diffuse large B cell lymphoma and could successfully detect the primary even though lymphoma lesions, unlike other tumors, supposedly lack fibrosis [22]. No dedicated study on FAP-specific PET for target volume delineation in upper gastrointestinal tract tumors could be found. 

### 2.7. Liver, Gallbladder, Pancreas

Shi et al. investigated the role of FAP-specific PET for primary tumors and metastases of the liver [26]. Seven of eight hepatocellular carcinomas showed at least moderate enhancement. Even though the cirrhotic livers showed an increased uptake per se; the tumors could be detected with high sensitivity. Giesel et al. noted a significantly reduced background activity of FAP-specific PET compared to FDG and therefore facilitated delineation of metastases, e.g., from pancreatic primaries which showed strong tracer uptake in general [37]. No dedicated study on FAP-specific PET for target volume delineation in these tumors could be found. 

### 2.8. Lower Gastrointestinal Tract

Koerber et al. conducted a study on radiotherapy planning for FDG-specific PET in 24 patients with tumors of the lower gastrointestinal tract (7 anal cancer, 4 rectal cancer, 6 sigmoid cancer, 5 colon cancer) [30]. The authors note a benefit for target volume delineation and a change in the oncological management of the patients in the majority of cases. Especially anal cancer, an entity where radiotherapy is a mainstay of definitive treatment, showed high uptake with FAP-specific PET.

### 2.9. Prostate

The experience of FAP-specific PET in prostate cancer is very limited. Kratochwil et al. conducted FAP-specific PET in PSMA negative tumors and found intermediate to high uptake [36]. A case report by Khreish et al. used FAP-specific PET in another patient with a PSMA-negative, highly dedifferentiated tumor adds to the results for this subgroup. No dedicated study on FAP-specific PET for target volume delineation in prostate cancer could be found. 

### 2.10. Bone

Chen et al report an improved detection sensitivity for bone metastasis compared to FDG-specific PET and a high tracer uptake of sarcomas [12]. Other studies also noted the propensity of FAP-specific PET to detect previously unknown osseous lesions [27]. No dedicated study on FAP-specific PET for target volume delineation in bone metastases could be found. 

## 3. Discussion

While early results of FAP-specific PET are available for a variety of different tumor entities, the results, albeit promising, should at this stage still be considered as hypothesis-generating. However, with the emergence of radiotherapy devices that integrate PET imaging into treatment planning and monitoring, the importance of biology-guided radiotherapy is likely to increase and could benefit greatly from advanced tracers [39]. 

The application of FAP-specific PET in the brain could benefit from the low background enhancement compared to FDG. While for many primary brain tumors such as gliomas, FDG is not the standard tracer anyway [40]; this could be of particular interest for the detection of brain metastases. If FAP-specific PET would be able to confidently detect brain metastases, this could enable extra- and intracranial staging with a single imaging modality, thereby saving the (in many cases) limited time patients with brain metastases have left and warranting further research in this direction. If size and location of brain metastases could be assessed with similar confidence as with MRI, FAP-specific PET could even be used for radiotherapy treatment planning, reducing the often unfortunately long time between the diagnosis of brain metastases that could be treated with stereotactic radiosurgery (SRS) and the next available slot for an additional MRI that satisfies the requirements for SRS planning. Expanding the use of FAP-specific PET for brain metastases would however require at least a head-to-head comparison with MRI prior to any PET-based treatment planning studies. 

For primary and secondary brain tumors, it still remains unclear if blood-brain-barrier disruption is a requirement for FAP-enhancement.

The possible reduction of staging and treatment planning imaging is also an incentive to investigate the use of FAP-specific PET in head and neck cancer. Following the trend of investigating a possible therapy de-escalation for subgroups with a favorable prognosis, accurately assessing the extent of tumor spread prior to treatment start could become even more important and enable radiation oncologists to reduce the area exposed to radiation and thereby reduce toxicities.

A reduction of the treatment field could also be achieved in lung tumors where differentiating tumor from normal tissue is often a difficult task, especially when the lung is affected by other conditions such as chronic obstructive pulmonary disease or idiopathic pulmonary fibrosis or if it has been treated with radiation.

An accurate assessment of tumor spread with FAP-specific PET and radiotherapy devices that can perform PET is especially promising for tumors of the upper gastrointestinal tract and the liver where conventional contrast-enhanced CT and FDG PET have several limitations and the variable positions of the organs in relation to one another often require substantial safety margins. This, in turn, exposes uninvolved organs to radiation, which can have a particularly detrimental effect in a group of patients that often already suffers from comorbidities and reduced performance status.

For the lower gastrointestinal tract, in general, the main benefit of FAP-specific PET was not limited to target volume delineation but a better assessment of tumor spread and, therefore, more information to decide the oncological management of a patient. However, even though nearly all head-to-head comparisons with FDG PET emphasize the increased sensitivity of FAP-specific PET, there is still not much information on its specificity, hinting at a decreased performance in patients who have other diseases that provoke or are defined by an inflammatory reaction such as chronic pancreatitis [25]. This also applies for the use of FAP-specific PET for bone metastases.

Possible limitations of the study include the presence of a patent COI in slightly more than half of the included publications. A possible limitation of the review is that the search was limited to the MEDLINE database which is, however, mitigated by the fact that FAP-specific tracers are currently only used by a limited amount of research groups whose results are published in MEDLINE-indexed journals. Another limitation is that the total number of patients who received FAP-specific PET for a given indication cannot be determined exactly. While some publications report that they analyze the same patient collective as another publication [23,35], it cannot be helped that some publications from the same group contain at least a subset of patients that is analyzed more than once.

As of July 2020, searching clinicaltrials.gov yielded eight recruiting or not yet recruiting prospective trials on FAP-specific PET, including several studies where resection and immunohistochemical staining will be performed afterwards in order to correlate tracer enhancement and the extent of tumor growth as well as the expression of FAP (Table 2). Furthermore, several studies will perform head-to-head comparisons with other tracers such as FDG, PSMA, FDOPA, and DOTATATE depending on the tumor entity. 

## 4. Methods

The review was conducted according to the PRISMA guidelines as applicable [41]. Studies published in English not earlier than 2018 that used FAP-specific PET in humans for any kind of cancer were included. No limits regarding the size of the patient collective or length of follow-up were applied. The MEDLINE database was searched on July 11th 2020 via the freely accessible PubMed interface. The query was designed to show results whose titles contained either of the words “fibroblast”, “FAP” or “FAPI” in combination with either “positron” or “PET” (example syntax: “((Fibroblast[Title]) OR (FAP[Title]) OR (FAPI[Title])) AND ((PET[Title]) OR (Positron[Title]))”). After exclusion of duplicates, the titles were screened and only relevant publications proceeded to full-text screening. All articles that did not focus on the use of FAP-specific PET in cancer patients and did not provide information on the ability of FAP-specific PET to detect and delineate tumors were excluded. Risk of bias in individual studies was assessed by gathering the conflict of interest (COI) with a concrete relation to the submitted work and funding statements as reported in each publication. 

## 5. Conclusions

In conclusion, the first studies on FAP-specific PET in cancer patients show promising results, especially when considering the advances in the field of radiation oncology in general. However, more head-to-head comparisons with existing imaging modalities, PET followed by histopathological examinations, larger studies, and ultimately randomized trials will be required before its definitive impact can be assessed.

## Figures and Tables

**Figure 1 cancers-12-02629-f001:**
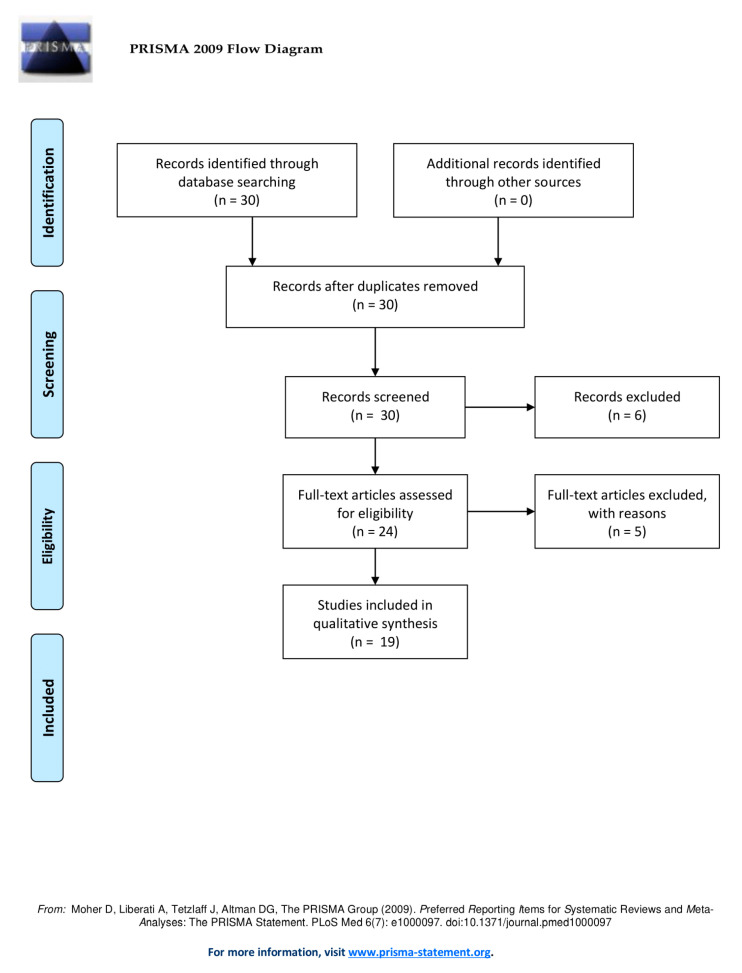
Workflow of the literature search according to PRISMA guidelines.

**Figure 2 cancers-12-02629-f002:**
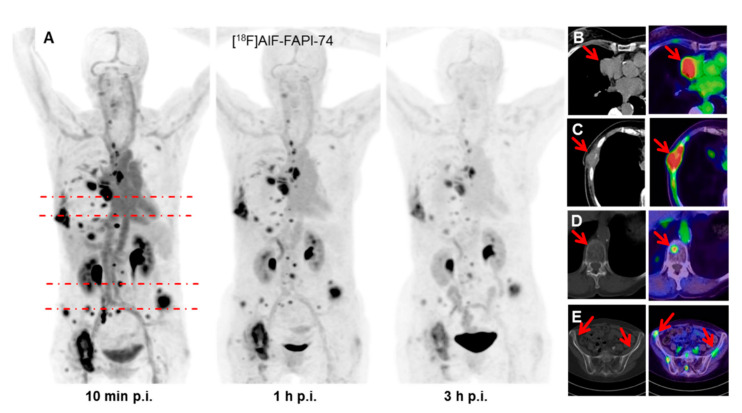
Maximum-intensity-projections of 18F-FAPI-74 PET at 10 min, 1 h and 3 h p.i. (**A**). FAPI- PET/CT presents favorable discrimination between tumor and myocardium (**B**). Some FAPI-positive lesions were confirmed by CT-correlate (**C**), while additional bone lesions were only detected per FAPI- PET (**D**,**E**). This research was originally published in JNM. Giesel, Adeberg et al. [24] FAPI-74 PET/CT using either 18F-AlF or cold-kit 68Ga-labeling: Biodistribution, Radiation Dosimetry and Tumor Delineation in Lung Cancer Patients. J Nucl Med. 2020. ^©^SNMMI.

**Table 1 cancers-12-02629-t001:** Summary of studies on FAP-specific PET in cancer patients. COI: Conflict of Interest.

Title	Author	Year	Tumor (Patients)	Tracer (Patients)	Key Findings	COI	Funding
^68^Ga-NOTA-FAPI-04 PET/CT in a patient with primary gastric diffuse large B cell lymphoma: comparisons with [(18)F] FDG PET/CT	Wang et al. [22]	2020	Gastric diffuse large B cell lymphoma (1 patient)	^68^Ga- FAPI-04 (1 patient)	FAP-specific PET showed considerable uptake in the primary gastric lymphoma and less uptake in regional lymph nodes compared to FDG	none	Beijing Municipal Science & Technology Commission
Fibroblast Activation Protein (FAP) specific PET for advanced target volume delineation in Glioblastoma	Windisch et al. [23] * same patient collective as Röhrich et al.	2020	Glioblastoma (13 patients)	^68^Ga-FAPI-02 (2 patients) ^68^Ga-FAPI-04 (11 patients)	Using an SUV threshold of 7-fold the background in healthy brain tissue as a threshold, resulted in tumor volumes equal to that on T1-weighted contrast-enhanced MRI Radiotherapy target volumes based on MRI and FAP-specific PET were incongruent	Patent application	none
FAPI-74 PET/CT Using Either ^18^F-AlF or Cold-kit ^68^Ga-labeling: Biodistribution, Radiation Dosimetry and Tumor Delineation in Lung Cancer Patients	Giesel, Adeberg et al. [24]	2020	Non small cell lung cancer (10 patients) Adenocarcinoma (8 patients) Squamous cell carcinoma (2 patients)	^18^F-FAPI-74 (10 patients)	No difference between adenocarcinoma and SCC Lower radiation burden compared to ^18^F-FDG Best correlation to CT-based target volumes occurred at an SUV of 3-fold the background	Patent application, Shares in Consultancy group	none
Usefulness of [^68^Ga] Ga-DOTA-FAPI-04 PET/CT in patients presenting with inconclusive [^18^F] FDG PET/CT findings	Chen et al. [25]	2020	Gastric cancer (12 patients) Signet ring cell carcinoma (4 patients) Adenocarcinoma (8 patients) Non small cell lung cancer (10 patients) Adenocarcinoma (10 patients) Small cell lung cancer (1 patient) Liver cancer (9 patients) Hepatocellular carcinoma (5 patients) Intrahepatic cholangiocarcinoma (4 patients) Nasopharyngeal cancer (7 patients) c (7 patients) Esophageal cancer (4 patients) Squamous cell carcinoma (4 patients) Breast cancer (4 patients) Invasive ductal carcinoma (3 patients) Invasive lobular carcinoma (1 patient) Cholangiocarcinoma (4 patients) Adenocarcinoma (4 patients) Ovarian cancer (2 patients) High-grade serous adenocarcinoma (2 patients) Cancer of unknown primary (2 patients) Squamous cell carcinoma (2 patients) Pancreatic cancer (1 patient) Adenocarcinoma (1 patient) Cervical cancer (1 patient) Squamous cell carcinoma (1 patient) Renal cancer (1 patient) Clear cell carcinoma (1 patient) Appendiceal carcinoma (1 patient) Mucinous adenocarcinoma (1 patient)	^68^Ga-FAPI-04 (59 patients)	Lower absolute signal of FAP-specific PET for brain metastases but higher tumor-to-background ratio compared to FDG FAP-specific PET identified more lesions than FDG-PET especially in peritoneal carcinomatosis, liver and skeletal metastases that in some cases lead to changes to staging and treatment Good tumor-to background ratio of FAP-specific PET in liver and gastric cancer The value of FAP-specific PET for differentiation of benign from malignant lesions requires further investigation due to possible false positives associated with inflammation-induced fibrosis	none	National Natural Science Foundation of China
Fibroblast imaging of hepatic carcinoma with ^68^Ga-FAPI-04 PET/CT: a pilot study in patients with suspected hepatic nodules.	Shi et al. [26]	2020	Liver cancer (13 patients) Hepatocellular carcinoma (11 patients) Intrahepatic cholangiocarcinoma (2 patients) Gastric cancer (1 patient, liver metastases) Adenocarcinoma (1 patient) Breast cancer (1 patient, liver metastases) Sigmoid carcinoma (1 patient, liver metastases) Adenocarcinoma (1 patient)	^68^Ga-FAPI-04 (16 patients)	High sensitivity of FAP-specific PET for detection of HCC and ICC Hepatic uptake was correlated with cirrhosis and hepatitis	none	National Natural Science Foundation of China, Fundamental Research Funds for the Central Universities, CAMS Innovation Fund for Medical Sciences
^68^Ga-FAPI PET/CT Detects Gastric Signet-Ring Cell Carcinoma in a Patient Previously Treated for Prostate Cancer	Pang et al. [27]	2020	Gastric cancer (1 patient) Signet ring cell carcinoma (1 patient)	Not specified (1 patient)	FAP-specific PET visualized the primary tumor and metastases Tracer uptake also occurred in both adrenal glands	none	none
Fibroblast activation protein inhibitor (FAPI) PET for diagnostics and advanced targeted radiotherapy in head and neck cancers	Syed et al. [8]	2020	Head and neck (14 patients) Squamous cell carcinoma (12 patients) Mucoepidermoid carcinoma (1 patient) Undifferentiated (1 patient)	Not specified (14 patients)	Using an SUV of 5-fold the threshold resulted in tumor volumes most similar to CT Tumor volumes based on FAP-specific PET and CT were incongruent and in some cases GTVs based on FAP-specific PET were not covered by PTVs based on CT	Patent application, Shares in Consultancy group	Open Access funding provided by Projekt DEAL
Comparison of ^68^Ga-FAPI and ^18^F-FDG PET/CT in a Patient With Cholangiocellular Carcinoma: A Case Report	Pang et al. [28]	2020	Cholangiocarcinoma (1 patient)	Not specified (1 patient)	FAP-specific PET detected more metastatic lesions than FDG PET in one patient with cholangiocarcinoma	none	none
FAP-specific PET signaling shows a moderately positive correlation with relative CBV and no correlation with ADC in 13 IDH wildtype glioblastomas	Röhrich et al. [11] * same patient collective as Windisch et al.	2020	Glioblastoma (13 patients)	^68^Ga-FAPI-02 (2 patients) ^68^Ga-FAPI-04 (11 patients) ∙	Intensity of FAP-specific PET in glioblastoma does most likely not reflect cell density but expression of FAP Intensity of FAP-specific PET does not solely reflect perfusion but is not completely independent from perfusion either	Patent application	Federal Ministry of Education and Research
Comparison of [^68^Ga] Ga-DOTA-FAPI-04 and [^18^F] FDG PET/CT for the diagnosis of primary and metastatic lesions in patients with various types of cancer	Chen et al. [12]	2020	Non small cell lung cancer (12 patients) Adenocarcinoma (11 patients) Adenosquamous carcinoma (1 patient) Small cell lung cancer (2 patients) Liver cancer (11 patients) Hepatocellular carcinoma (6 patients) Intrahepatic cholangiocarcinoma (5 patients) Nasopharyngeal cancer (7 patients) Nonkeratinizing, undifferentiated carcinoma (7 patients) Gastric cancer (8 patients) Adenocarcinoma (7 patients) Signet ring cell carcinoma (1 patient) Pancreatic cancer (4 patients) Adenocarcinoma (3 patients) Adenosquamous carcinoma (1 patient) Esophageal cancer (5 patients) Squamous cell carcinoma (5 patients) Glioma (4 patients) Glioblastoma (2 patients) Grade II glioma (1 patient) Grade III glioma (1 patient) Ovarian cancer (6 patients) High-grade serous carcinoma (6 patients) Colorectal cancer (8 patients) Adenocarcinoma (8 patients) Cervical cancer 3 patients) Squamous cell carcinoma (3 patients) Sarcoma (3 patients) Osteosarcoma (1 patient) Hemangiosarcoma (1 patient) Liposarcoma (1 patient) Neuroendocrine tumor (3 patients) G2 (1 patient) G3 (2 patients) Breast cancer (1 patient) Invasive ductal carcinoma (1 patient)	^68^Ga-FAPI-04 (75 patients)	FAP-specific PET showed particular good tumor-to-background ratios compared to FDG for hepatic and peritoneal tumor manifestations FAP-specific PET showed higher sensitivity in the detection of lymphonodal, osseous and visceral metastases with no difference in specificity	none	National Natural Science Foundation of China
^68^Ga-FAPI PET/CT Improves Therapeutic Strategy by Detecting a Second Primary Malignancy in a Patient With Rectal Cancer.	Chen et al. [29]	2020	Rectal cancer (1 patient) Non small cell lung cancer (1 patient) Adenocarcinoma (1 patient)	Not specified (1 patient)	FAP-specific PET was successfully used for biopsy planning in a lung node that had moderate uptake on FDG PET	none	none
The role of FAPI-PET/CT for patients with malignancies of the lower gastrointestinal tract - first clinical experience	Koerber et al. [30]	2020	Anal cancer (7 patients) Rectal cancer (4 patients) Sigmoid cancer (6 patients) Colon cancer (5 patients)	^68^Ga-FAPI-04 (16 patients) ^68^Ga-FAPI-46 (6 patients)	FAP-specific PET changed the classification according to TNM in 50% of treatment naïve patients FAP-specific PET caused a change in the (radio)oncological management in 81% of patients	Patent application	none
Intense FAPI Uptake in Inflammation May Mask the Tumor Activity of Pancreatic Cancer in ^68^Ga-FAPI PET/CT	Luo et al. [31]	2020	Pancreatic cancer (1 patient)	Not specified (1 patient)	FAP-specific PET, unlike FDG-PET, did not delineate a mass that turned out to be pancreatic cancer most likely due to surrounding, tumor-induced pancreatitis	none	CAMS Initiative for Innovative Medicine
Radiation dosimetry and biodistribution of ^68^Ga-FAPI-46 PET imaging in cancer patients	Meyer et al. [32]	2019	Breast cancer (1 patient) Gastric cancer (1 patient) Head and neck cancer (1 patient) Oropharynx carcinoma (1 patient) Pancreatic cancer (1 patient) Cholangiocarcinoma (1 patient)	^68^Ga-FAPI-46 (6 patients)	FAP-specific PET showed high tumor-to-background ratios that increased over the three timepoints	Patent application	none
Positive FAPI-PET/CT in a metastatic castration-resistant prostate cancer patient with PSMA-negative/FDG-positive disease	Khreish et al. [33]	2019	Prostate cancer (1 patient)	^68^Ga- FAPI-04 (1 patient)	FAP-specific PET showed intense uptake of all metastases of a patient with dedifferentiated, advanced-stage prostate cancer	Patent application	none
IDH-wildtype glioblastomas and grade III/IV IDH-mutant gliomas show elevated tracer uptake in fibroblast activation protein-specific PET/CT	Röhrich et al. [34] * same patient collective as Windisch et al.	2019	Glioma (18 patients) Grade II glioma (3 patients) Grade III glioma (1 patient) Glioblastoma (14 patients)	^68^Ga-FAPI-02 (2 patients) ^68^Ga-FAPI-04 (16 patients)	FAP-specific PET showed high tracer accumulation in grade III/IV but not grade II gliomas	none	Federal Ministry of Education and Research
FAPI-PET/CT improves staging in a lung cancer patient with cerebral metastasis	Giesel et al. [35]	2019	Non small cell lung cancer (1 patient) Adenocarcinoma (1 patient)	^68^Ga- FAPI-04 (1 patient)	FAP-specific PET detected two brain metastases (>= 8 mm)	Patent application	
^68^Ga-FAPI PET/CT: Tracer Uptake in 28 Different Kinds of Cancer.	Kratochwil et al. [36]	2019	Renal cancer (1 lesion) Insulinoma (1 lesion) Prostate cancer (16 lesions) Neuroendocrine differentiation (3 lesions) Adenocarcinoma (13 lesions Thyroid cancer (18 lesions) Differentiated (12 lesions) Medullary (6 lesions) Pheochromocytoma (4 lesions) Adenoid cycstic carcinoma (4 lesions) Gastric cancer (3 lesions) Liver cancer (5 lesions) Hepatocellular carcinoma (5 lesions) Cervical cancer (3 lesions) Small intestine cancer (6 lesions) Neuroendocrine tumors (3 lesions Anal cancer (3 lesions) Colorectal cancer (38 lesions) Chordoma (1 lesion) Desmoid (4 lesions) Ovarian cancer (8 lesions) Head and neck cancer (34 lesions) Thymus cancer (4 lesions) Pancreatic cancer (51 lesions) Lung cancer (25 lesions) Breast cancer (12 lesions) Cholangiocellular carcinoma (12 lesions) Esophageal cancer (6 lesions) Salivary gland cancer (6 lesions) Sarcoma (8 lesions) Cancer of unknown primary (7 lesions)	^68^Ga- FAPI-04 (80 patients)	The highest uptake was observed in lung cancer, breast cancer, esophageal cancer as well as cholangiocellular carcinoma and sarcoma The tracer showed low peritoneal uptake which facilitated diagnosis of peritoneal carcinomatosis FAP-specific PET had limitations similar to those of FDG-PET for renal cell carcinoma, pheochromocytoma and thyroid cancer.	Patent application	none
^68^Ga-FAPI PET/CT: Biodistribution and Preliminary Dosimetry Estimate of 2 DOTA-Containing FAP-Targeting Agents in Patients with Various Cancers	Giesel et al. [37]	2018	Breast cancer (2 patients) Colorectal cancer (4 patients) Cancer of unknown primary (2 patients) Head and neck cancer (8 patients) Liver cancer (2 patients) Hepatocellular carcinoma (2 patients) Sarcoma (1 patient) Liposarcoma (1 patient) Non small cell lung cancer (5 patients) Esophageal cancer (2 patients) Pancreatic cancer (13 patients) Prostate cancer (4 patients) Renal cancer (1 patient) Thyroid cancer (3 patients) Uterus cancer (1 patient) Ovarian cancer (1 patient)	^68^Ga-FAPI-02 (25 patients) ^68^Ga- FAPI-04 (24 patients)	Tumor-to-background ratios of FAP-specific PET were at least similar to FDG-PET	Patent application	none

**Table 2 cancers-12-02629-t002:** Recruiting and not-yet-recruiting trials for FAP-specific PET on clinicaltrials.gov.

Title	Estimated Enrollment	Estimated Study Completion Date	Tumors	Tracer	Location	Key Interventions
The Role of ^68^Ga-FAPI-04 PET/CT in Gastric and Pancreatic Cancers	25 patients	06/2021	Pancreatic Cancer Gastric Cancer	^68^Ga-FAPI-04	Tel-Aviv Sourasky Medical Center	Head-to-head comparison between FAP-specific PET and FDG PET (max. interval: 10 days) Immunohistochemical staining of FAP in removed tumors for correlation with imaging
^68^Ga-FAPI PET Imaging in Malignancy	30 patients	12/2021	Various	^68^Ga-FAPI (not specified)	Stanford University Hospitals and Clinics	FAP-specific PET will be assessed for feasibility on a Likert scale from 1 (non-diagnostic) to 5 (excellent)
Comparison of FDG and FAPI in Patients With Various Types of Cancer	600 patients	12/2021	Various	^68^Ga-FAPI-04	The First Affiliated Hospital of Xiamen University	Head-to-head comparison between FAP-specific PET and FDG PET
PET Biodistribution Study of 68Ga-FAPI-46 in Patients With Prostate Cancer: A Prospective Exploratory Biodistribution Study With Histopathology Validation	30 patients	07/2023	Prostate Cancer	^68^Ga-FAPI-46	UCLA/Jonsson Comprehensive Cancer Center	Assessment of biodistribution Immunohistochemical staining of FAP in removed tumors for correlation with imaging Comparison between FAP-specific PET and previously performed PSMA PET (max. interval: 3 months)
PET Biodistribution Study of ^68^Ga-PSMA-11 and ^68^Ga-FAPI-46 in Patients With Non-Prostate Cancers: An Exploratory Biodistribution Study With Histopathology Validation	30 patients	10/2021	Various	^68^Ga-FAPI-46	UCLA/Jonsson Comprehensive Cancer Center	Assessment of biodistribution Immunohistochemical staining of FAP in removed tumors for correlation with imaging Optional head-to-head comparison with PSMA PET
PET Biodistribution Study of 68Ga-FAPI-46 in Patients With Different Malignancies: An Exploratory Biodistribution Study With Histopathology Validation	30 patients	07/2024	Various	^68^Ga-FAPI-46	UCLA/Jonsson Comprehensive Cancer Center	Assessment of biodistribution Immunohistochemical staining of FAP in removed tumors for correlation with imaging Comparison between FAP-specific PET and previously performed FDG/DOTATE/FDOPA or other PET (max. interval: 3 months)
Positron Nuclide Labeled NOTA-FAPI PET Study in Lymphoma	200 patients	12/2020	Lymphoma	^68^Ga-FAPI-04^18^F-FAPI-04	Peking University Cancer Hospital	Head-to-head comparison between FAP-specific PET and FDG PET (max. interval: 10 days)
PET Biodistribution Study of ^68^Ga-FAPI-46 in Patients With Sarcoma: An Exploratory Biodistribution Study With Histopathology Validation	30 patients	07/2024	Sarcoma	^68^Ga-FAPI-46	UCLA/Jonsson Comprehensive Cancer Center	Assessment of biodistribution Immunohistochemical staining of FAP in removed tumors for correlation with imaging Comparison between FAP-specific PET and previously performed FDG PET (max. interval: 3 months)

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
