# Peer review of "Clinical Results of Fibroblast Activation Protein (FAP) Specific PET and Implications for Radiotherapy Planning: Systematic Review"

_cancers, 2020, doi:10.3390/cancers12092629_

Round 1

Reviewer 1 Report

The article is a systematic review (19 articles) of clinical use of Fibroblast Activation Protein (FAP) specific PET in multiple malignancies (>30).

A special emphasis for its possible role for assessing essential "target volumes" in radiotherapy planning has  been presented (in the title).

This article is planned to be part of tumor microenvironment section: however, it is not explained (not even speculated) how FAP should act is target determination in tumor microenvironment. This is mandatory, because there are different cancer types ..., and FAP may play a different role in different tumor stromal types

FAP has been used as target in clinical setting since early 1990´s  (but the forgotten for some years). The reappearance is due to power of PET imaging. This article is about PET imaging, but there are no images. A couple of images would be nice (for the readers of Cancers). The authors could also list briefly the small molecules in clinical use.

The Systematic analysis (PRISMA guidelines) is appropriate and the listing of  current trials is relevant.

Thus, the publication of this systematic review is strongly supported.

For improvement of this article: The mechanism of FAP targeting in the tumor microenvironment should be explained in a more detailed manner (intro or discussion), similarly clinical in vivo targeting (own data/ or from articles) should be shown (eg. PET images) and finally target volume definitions should at least be speculated  (what is measured? and where it  helps radiation therapy planning?).

Author Response

Dear reviewer, 

Thank you for your in-depth review of our commentary as well as the positive assessment of our work and its purpose. Your suggestions to further strengthen it are highly appreciated and have caused the following changes and additions:

Reviewer: This article is planned to be part of tumor microenvironment section: however, it is not explained (not even speculated) how FAP should act is target determination in tumor microenvironment. This is mandatory, because there are different cancer types ..., and FAP may play a different role in different tumor stromal types

Response: We and the editors agree with your assessment of the lack of discussion regarding the tumor microenvironment. The article has therefore been moved from the tumor microenvironment section to the section ‘cancer clinical trials’. In addition, we have extended the paragraph on preclinical findings in the introduction:

“A novel group of tracers that has emerged recently are substances targeting the fibroblast activation protein (FAP) on the surface of fibroblasts in the tumor stroma, so called cancer associated fibroblasts (CAFs) first published in 2018 [4]. As FAP is highly specific to a large subset of CAFs, early molecules targeting FAP have initially been designed to inhibit FAP and to thereby serve as potential therapeutic agents before being used as a PET tracer in the past [5,6]. Even though fibroblasts occur ubiquitously throughout the body, they normally express dipeptidyl peptidase 4 (DPP4) instead of the related FAP which is why preclinical in vitro and in vivo studies, FAP-specific tracers showed high specificity, affinity and rapid internalization [4].

Reviewer: FAP has been used as target in clinical setting since early 1990´s  (but the forgotten for some years). The reappearance is due to power of PET imaging. This article is about PET imaging, but there are no images. A couple of images would be nice (for the readers of Cancers). The authors could also list briefly the small molecules in clinical use.

Response: We have, in the meantime, obtained permission to re-utilize a figure from the Journal of Nuclear Medicine and included it in the revised manuscript. The small molecules that we have found are listed in the tracer section of the table (FAPI-02,-04,-74,-46).

Reviewer: The mechanism of FAP targeting in the tumor microenvironment should be explained in a more detailed manner (intro or discussion), similarly clinical in vivo targeting (own data/ or from articles) should be shown (eg. PET images) and finally target volume definitions should at least be speculated  (what is measured? and where it  helps radiation therapy planning?).

Response: For the added information on FAP in preclinical studies and the addition of an image see above. The possible use of FAP-specific PET for target volume delineation lies in the accurate depiction of tumor spread which is included for different entities in the discussion. While aspects such as dose painting will certainly be of interest in the future, we decided to not discuss them considering that this is still an area of controversy even for far more established PET tracers. 

Thank you for your contribution.

Reviewer 2 Report

Dear authors,

you present a thorough review of FAP specific PET and its implications for RT Planning. Your review is well planned and your exclusion criteria are reasonable as you excluded studies on benign disease and in vitro studies. However, even if you gave a review about clinical applications of FAP specific PET, I think it could increase readability and intelligibility if you include a short section about preclinical, in vitro findings, too. 

Furthermore you state in the Discussion section that another advantage of FAP specific PET might be that it reduces the waiting time for patients before SBRT of brain metastasis to receive a high quality MRI scan. In my opinion waiting time for PET scans are even longer than for MRI scans and this is not really a good argument for FAP specific PET. Please clarify.

Otherwise I fully recommend publications of this review.

Author Response

Dear reviewer, 

Thank you for your in-depth review of our commentary as well as the positive assessment of our work and its purpose. Your suggestions to further strengthen it are highly appreciated and have caused the following changes and additions:

Reviewer: you present a thorough review of FAP specific PET and its implications for RT Planning. Your review is well planned and your exclusion criteria are reasonable as you excluded studies on benign disease and in vitro studies. However, even if you gave a review about clinical applications of FAP specific PET, I think it could increase readability and intelligibility if you include a short section about preclinical, in vitro findings, too. 

Response: We have extended the section on preclinical findings in the introduction:

“A novel group of tracers that has emerged recently are substances targeting the fibroblast activation protein (FAP) on the surface of fibroblasts in the tumor stroma, so called cancer associated fibroblasts (CAFs) first published in 2018 [4]. As FAP is highly specific to a large subset of CAFs, early molecules targeting FAP have initially been designed to inhibit FAP and to thereby serve as potential therapeutic agents before being used as a PET tracer in the past [5,6]. Even though fibroblasts occur ubiquitously throughout the body, they normally express dipeptidyl peptidase 4 (DPP4) instead of the related FAP which is why preclinical in vitro and in vivo studies, FAP-specific tracers showed high specificity, affinity and rapid internalization [4].

Reviewer: Furthermore you state in the Discussion section that another advantage of FAP specific PET might be that it reduces the waiting time for patients before SBRT of brain metastasis to receive a high quality MRI scan. In my opinion waiting time for PET scans are even longer than for MRI scans and this is not really a good argument for FAP specific PET. Please clarify.

Response: We agree with your observations regarding the waiting time for PET/MRI and have changed this passage to avoid confusion. While MRIs are routinely conducted for the assessment of intracranial disease, a staging MRI in many cases does not fulfill the requirements for stereotactic radiosurgery planning in terms of slice thickness and available sequences as they are in most cases not ordered by a radiation oncologist. Oftentimes, an additional, dedicated SRS-MRI is ordered for treatment planning and this second MRI delays the treatment. If only one imaging study could be performed for intra- as well as extracranial staging and SRS planning, this would result in a reduction of the time to treatment. 

“If size and location of brain metastases could be assessed with similar confidence as with MRI, FAP-specific PET could even be used for radiotherapy treatment planning, reducing the often unfortunately long time between the diagnosis of a brain metastases that could be treated with stereotactic radiosurgery (SRS) and the next available slot for an additional MRI that satisfies the requirements for SRS planning.“

Thank you for your contribution.